# Microstructural differences in the osteochondral unit of terrestrial and aquatic mammals

Irina AD Mancini[1,2], Riccardo Levato[1,2,3†], Marlena M Ksiezarczyk[2,3†], Miguel Dias Castilho[2,3,4], Michael Chen[5], Mattie HP van Rijen[2,3], Lonneke L IJsseldijk[6], Marja Kik[6], P René van Weeren[1,2], Jos Malda[1,2,3]*

[1]Department of Clinical Sciences, Faculty of Veterinary Medicine, Utrecht University, Utrecht, Netherlands; [2]Regenerative Medicine Utrecht, Utrecht University, Utrecht, Netherlands; [3]Department of Orthopedics, University Medical Centre Utrecht, Utrecht, Netherlands; [4]Department of Biomedical Engineering, Eindhoven University of Technology, Eindhoven, Netherlands; [5]Department of Mathematical Sciences, University of Adelaide, Adelaide, Australia; [6]Division of Pathology, Department of Biomolecular Health Sciences, Faculty of Veterinary Medicine, Utrecht University, Utrecht, Netherlands

**\*For correspondence:**
j.malda@umcutrecht.nl

†These authors contributed equally to this work

**Competing interest:** The authors declare that no competing interests exist.

**Abstract** During evolution, animals have returned from land to water, adapting with morphological modifications to life in an aquatic environment. We compared the osteochondral units of the humeral head of marine and terrestrial mammals across species spanning a wide range of body weights, focusing on microstructural organization and biomechanical performance. Aquatic mammals feature cartilage with essentially random collagen fiber configuration, lacking the depth-dependent, arcade-like organization characteristic of terrestrial mammalian species. They have a less stiff articular cartilage at equilibrium with a significantly lower peak modulus, and at the osteochondral interface do not have a calcified cartilage layer, displaying only a thin, highly porous subchondral bone plate. This totally different constitution of the osteochondral unit in aquatic mammals reflects that accommodation of loading is the primordial function of the osteochondral unit. Recognizing the crucial importance of the microarchitecture-function relationship is pivotal for understanding articular biology and, hence, for the development of durable functional regenerative approaches for treatment of joint damage, which are thus far lacking.

## Editor's evaluation

It is important to determine microstructure-function relationships among different animal species, such as terrestrial and aquatic mammals, since it will help us understand articular biology and inform disease treatment. The authors compared the microstructure and biomechanical property of the osteochondral bone of the humeral head and found the cartilage of aquatic animals have a less stiff cartilage, a more random alignment of collagen fibers, and a lack of a calcified cartilage layer at the cartilage-bone interface. The specific composition of the osteochondral bone in aquatic mammals also reflects the changes in loading.

## Introduction

In the course of evolution, mammals have gone from land to water at several occasions, resulting in separate lineages ranging from Cetacea, which include whales and dolphins, to diverging lines, such

as the polar bear (*Ursus maritimus*) and sea otter (*Enhydra lutris*) (*Uhen, 2007*). Polar bears and sea otters have maintained their terrestrial form despite spending a significant portion of their life in water, while other species, in their evolutionary process, have become permanent residents of the aquatic environment, undergoing considerable morphological adaptations to marine life (*Uhen, 2007*). This long adaptation process that started in the Eocene, is particularly evident in Cetacea (*Uhen, 2007*; *Reidenberg, 2007*), where thoracic limbs became flippers, hind limbs regressed and disappeared, and vertebral bones adopted a more uniform morphology (*Cozzi et al., 2010*; *Thewissen et al., 2006*).

The main drive for these changes was obviously meeting the physical demands of life in an aquatic environment, where locomotion is largely disentangled from weight-bearing (*Turnbull and Cowan, 1999*), and thus radically different than as experienced on land. In fact, in water, buoyancy changes the influence of gravity dramatically and water is denser and more viscous than air, influencing resistance (*Dejours, 1987*). Therefore, locomotion in water happens when the mechanism that propels the animal forward is powerful enough to counter the effects of drag. Different mammalian species specialized in the form of diverse morphologies of their body and limbs (or fins), however, always with the common denominator of the development of propulsive surfaces that oscillate to generate thrust. Swimming of aquatic mammals differs significantly in mechanical efficiency and stress distribution from locomotion in terrestrial mammals relying on limb movement. In particular, pinnipeds, cetaceans and sirenians descend from mammals that evolved to move limbs in a vertical plane, and consequently use a hydrodynamic lift-based momentum exchange to move through water (*Farnum, 2007*).

In all mammals, regardless whether they live on land or in water, locomotion is effectuated by the musculoskeletal system. Its framework is composed by the skeleton, whose bony components articulate via joints. The function of synovial joints in particular, is to ensure the correct articulation of adjacent bones. In addition, it minimizes friction between the bony components by providing a smooth articulating surface and lubrication to accommodate the biomechanical forces generated by locomotion, transmitting and dampening them (*Sophia Fox et al., 2009*). The fundamental element that enables physiological function of the synovial joint is the osteochondral unit, composed of articular cartilage and bone. In the front extremities, synovial joints are the principal joint type in terrestrial mammals; in aquatic mammals they have been preserved during evolution at the articulating ends of the humerus, radius, and ulna (*Sanchez and Berta, 2010*), but have been replaced by fibrous joints in the distal joints of flippers, where bony elements are connected by dense connective tissue (*Thewissen, 2009*).

Earlier comparative studies on the musculoskeletal system of terrestrial and aquatic mammals have largely focused on bone differences. This is probably because bone is a tissue known to respond extensively and quickly to changes in loading, as described by Wolff's law and Frost's mechanostat theory (*Wolff, 1893*; *Frost, 2001*). Studies on the macroscopic structures have shown that the bones of the cetacean arm and forearm adopted an hourglass-like form, and that the medullary cavity disappeared to strengthen flipper resistance for steering during swimming (*de Buffrénil and Schoevaert, 1988*; *Felts and Spurrell, 1965*). At tissue level, evolutionary adaptations of the bone have been described for both terrestrial (*Doube et al., 2010*; *Mancini et al., 2019*) and aquatic mammals (*Gray et al., 2007*), with a particular focus on how aquatic mammals manage buoyancy by variations in their structural bone density (*Gray et al., 2007*; *Taylor, 2000*). However, besides a few studies on some joint diseases (*Nganvongpanit et al., 2017*; *Turnbull and Cowan, 1999*), there is very limited knowledge of specific adaptations of the architecture of the cartilage extracellular matrix in aquatic mammals. Moreover, subchondral bone microstructure and related mechanical characteristics, which provide insight in essential aspects of the functionality of the osteochondral unit in joints, have not yet been studied in detail.

The osteochondral unit is the pivotal element of the joint, with a fundamental clinical relevance in highly prevalent joint disorders as osteoarthritis (OA), for which no effective, durable treatment exists yet (*Lepage et al., 2019*). In terrestrial mammals, comparative studies on the osteochondral unit have shown that the biochemical components of the tissues are strongly preserved across a wide range of species (*Malda et al., 2013*). A negative allometric relationship exists between cartilage thickness and body mass (*Malda et al., 2013*) and it is the trabecular bone structure that adapts to increasing body mass (*Mancini et al., 2019*). At present, comparative studies of the osteochondral unit of terrestrial

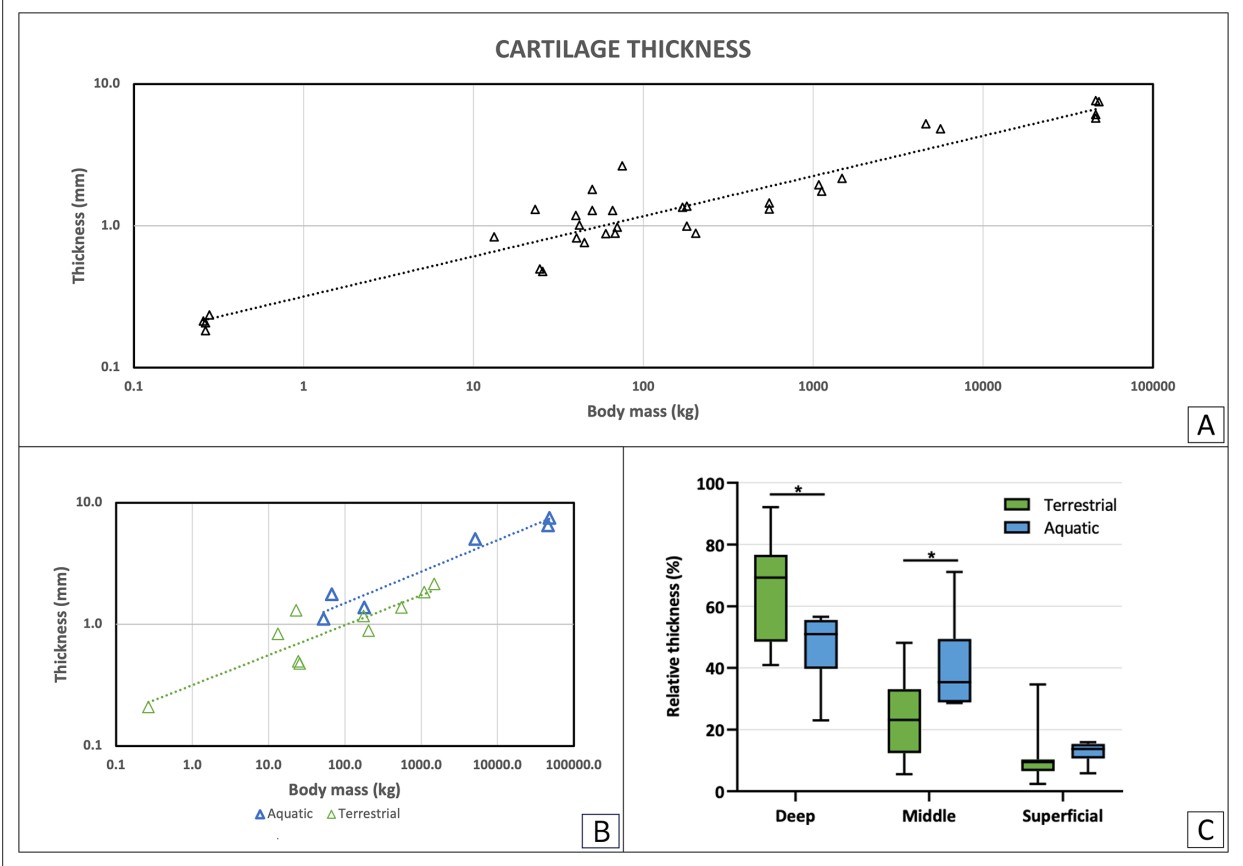

**Figure 1.** Cartilage thickness as analyzed from histological sections. (**A**) Total articular cartilage thickness correlates with body mass with a negative allometric relationship ($R^2$=0.91, a=0.28). (**B**) Total cartilage thickness in correlation with body mass for terrestrial and aquatic mammals separately. In green are terrestrial mammals ($R^2$=0.80, a=0.26), in blue are aquatic mammals ($R^2$=0.96, a=0.26). (**C**) Average of relative layer thickness (%) was calculated for each cartilage layer in terrestrial (green) and aquatic (blue) mammals. Comparison shows a significant difference between the two groups in both the deep and middle layers ($p < 0.01$).

The online version of this article includes the following source data for figure 1:

**Source data 1.** Details of the cartilage samples from the aquatic and terrestrial mammals.

and aquatic mammalian species do not exist, despite the high clinical relevance of the structure and the potential fundamental insights such studies would provide.

The aim of this study was, therefore, to investigate if and how the structure of both the cartilage and the bone component of the osteochondral unit differs between mammals living on land or in water. Given that the biological function of the osteochondral unit is largely structural in nature and determined by the composition and the architecture of the extracellular matrix (ECM) of both the cartilage and the bone part, the present work focused on analyzing the microstructural composition and architectural features of the osteochondral units from the humeral head of six aquatic and nine terrestrial mammalian species.

## Results

### Histological and polarized light microscopy analysis of cartilage

Articular cartilage thickness varied widely between species, ranging from 209 µm in the rat to 7660 µm in the sperm whale (*Physeter macrocephalus*). Total cartilage thickness correlated with body mass (BM) with a negative allometric relationship ($R^2$=0.91, a=0.28, *Figure 1A*). This relationship was maintained when assessing separately the terrestrial ($R^2$=0.80, a=0.26, green, *Figure 1B*) and aquatic mammals ($R^2$=0.96, a=0.26, blue, *Figure 1B*). The relative thickness of the superficial, middle and deep layers of the cartilage (expressed as percentage of total thickness) in terrestrial and aquatic mammals was

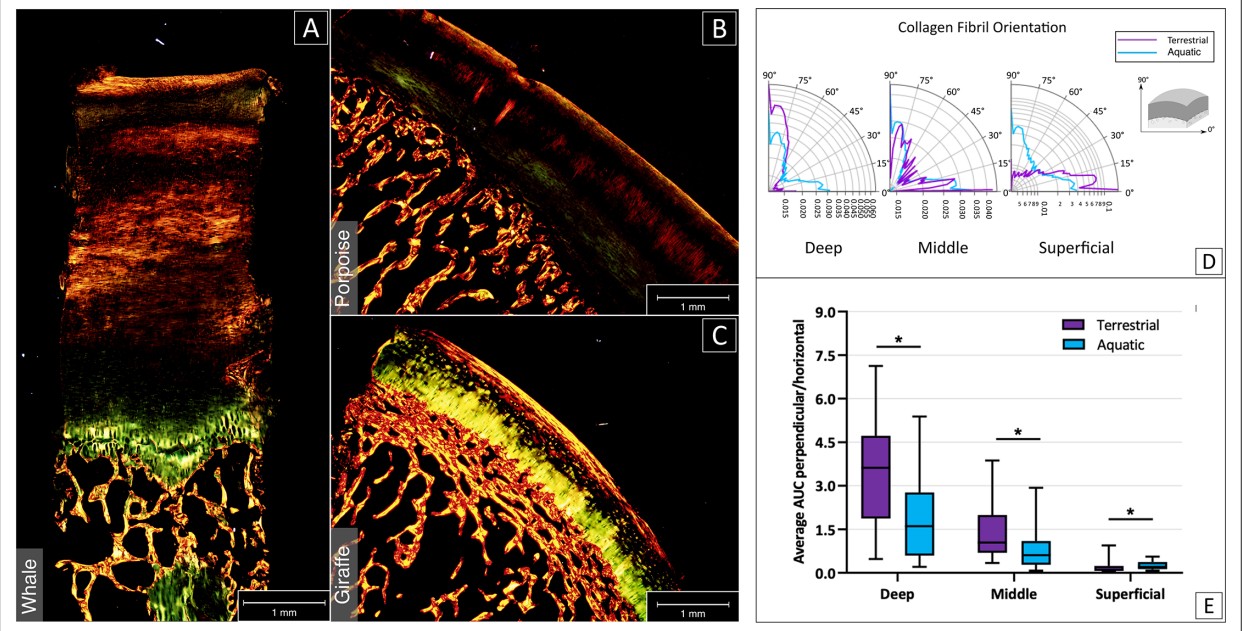

**Figure 2.** Analysis of the collagen fibre orientation based on picrosirius red staining and imaging with PLM. The collagen fibers orientation of aquatic mammals (sperm whale, **A**; harbor porpoise; **B**) appeared less organized than fiber orientation in terrestrial mammals (giraffe; **C**). (**C**) Cartilage structure in terrestrial mammals featured a clear distinction between deep, middle and superficial layer (respectively in yellow, black and red/yellow from bone to surface). (**D**) Polar graphical representation of collagen fiber distribution for deep, middle and superficial layer in terrestrial (violet) and aquatic (cerulean) mammals. 0° are fibers parallel to the surface, 90° are fibers perpendicular to the surface. (**E**) Average peak area was calculated for each cartilage layer in both terrestrial (violet) and aquatic (cerulean) mammals, showing significant differences between the two groups in all layers (p<0.05).

compared. The analysis showed significant differences in relative layer thickness, with the deep layer being relatively thicker in terrestrial than in aquatic mammals, and the middle layer being relatively thicker in aquatic than in terrestrial mammals. No significant differences were found in superficial layer relative thickness (*Figure 1C*).

Polarized Light Microscopy (PLM) enabled the visualization of the different collagen fiber orientations (*Figure 2A–C*), revealing that in terrestrial mammals, the fibers are predominantly oriented perpendicularly to the surface in the deep layer and parallel to the surface in the superficial layer, whereas for the aquatic mammals this arcade orientation of fibers could not be distinguished (*Figure 2D–E*). However, some alignment in the superficial layer was also observed (*Figure 2B*), albeit not to the same extend as in the samples of terrestrial mammals.

## Biomechanical analysis

Characterization of biomechanical behavior showed a marked difference in the stress relaxation curve of aquatic and terrestrial cartilage samples (*Figure 3A*). Analysis of time dependent mechanical properties showed that peak modulus was significantly higher in terrestrial mammals than in aquatic mammals (*Figure 3C*, p<0.05), while at equilibrium, cartilage appeared to be only slightly stiffer in terrestrial mammals (*Figure 3D*). Initial stress relaxation appeared slower for aquatic mammals and steeper for terrestrial mammals (*Figure 3A*, *Figure 3—figure supplement 1*). No significant differences were observed in the phase of slow relaxation (*Figure 3—figure supplement 1*). In addition, comparison of the relationship of peak and equilibrium modulus and body mass showed, with this sample size, no significant differences (*Figure 3E and F*).

## Histological and micro-computed tomography analysis of the interface

Imaging through histology and micro-CT showed that in aquatic mammals (*Figure 4A and B*) the interface between the hyaline articular cartilage and the subchondral bone was lacking a layer of calcified cartilage and a dense subchondral plate, as is typically observed at the cartilage-bone interface in terrestrial mammals (*Figure 4C*; data of additional species provided in *Figure 4—figure supplement*

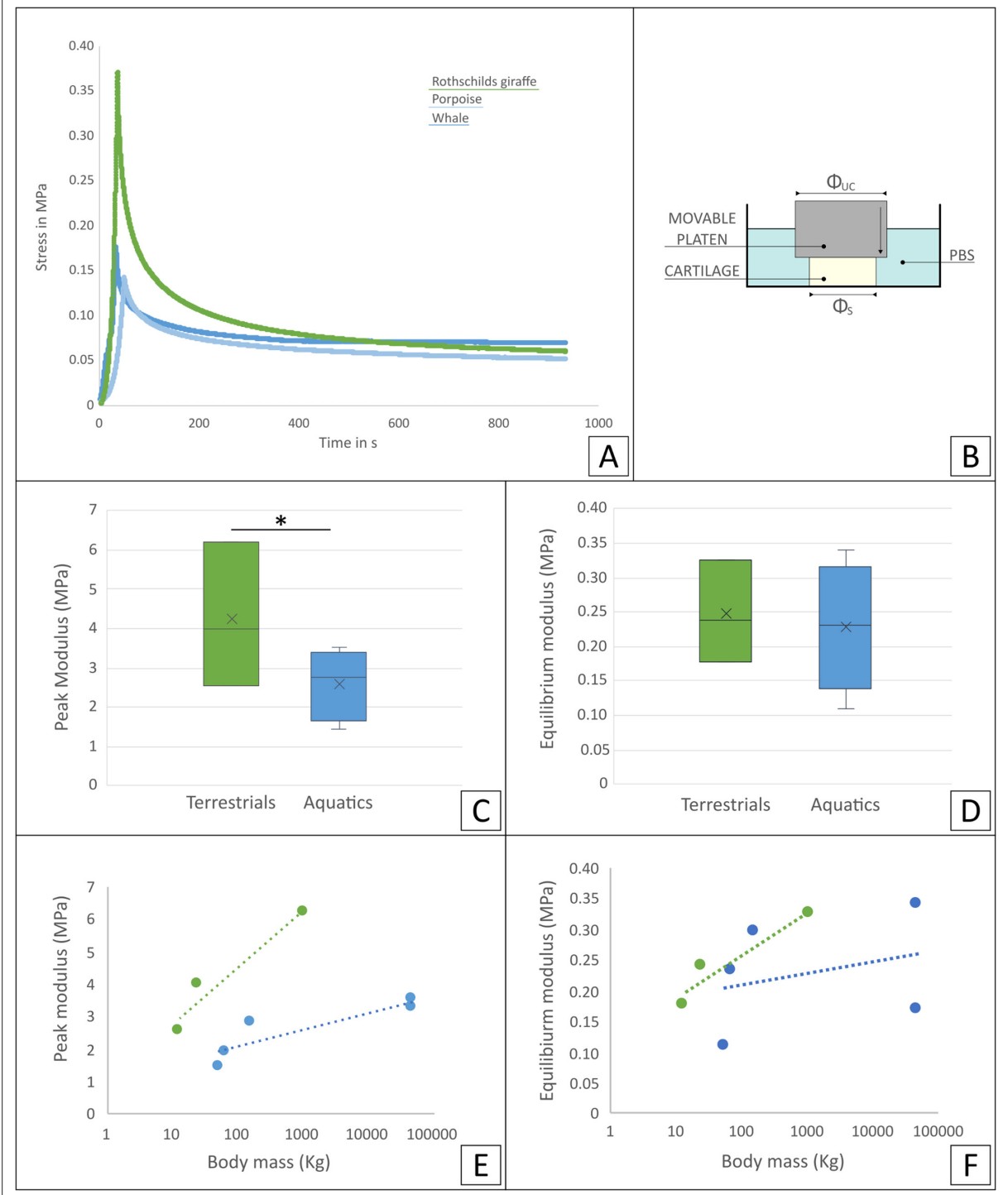

**Figure 3.** Mechanical analysis of the tissue samples. (**A**) Representative stress relaxation curve of some of the tested cartilage samples (Rothschilds giraffe, porpoise and sperm whale). (**B**) Schematics of loading methodology: unconfined compression geometry (congruent loading) used on cartilage samples immersed in a PBS bath. Comparison of time dependent mechanical properties: (**C**) peak and (**D**) equilibrium modulus for terrestrial (green) and aquatic mammals (blue). * Indicates a significant difference. Relationship of (**E**) peak and (**F**) equilibrium moduli average for each species with body mass for terrestrial (green) and aquatic mammals (blue).

The online version of this article includes the following figure supplement(s) for figure 3:

**Figure supplement 1.** Stress relaxation.

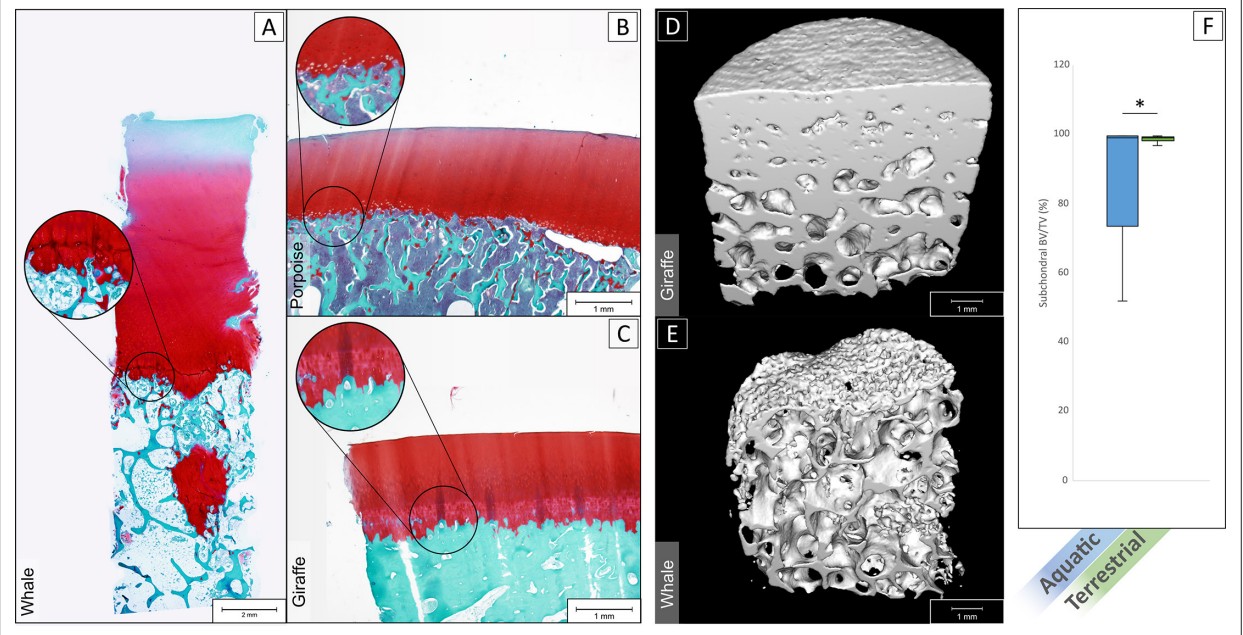

**Figure 4.** Histological staining and micro-CT analysis of the osteochondral tissues. Safranin-O staining of histological sections of the tissue of a sperm whale (**A**), harbor porpoise (**B**) and Rothschild's giraffe (**C**). Aquatic mammals (**A,B**) showed an abrupt transition from hyaline cartilage to subchondral trabecular bone. Terrestrial mammals (**C**) displayed the characteristic calcified cartilage layer with tidemark (in magnification bubble) in between and a dense subchondral plate before the subchondral trabecular bone. (**D**) 3D rendered micro-CT scan of giraffe bone core, showing on top a dense surface area that interfaces with the (calcified) cartilage (not shown in scan) and a gradually more porous structure as the subchondral tissue transitions to trabecular bone. (**E**) 3D rendered micro-CT scan of a whale bone core, showing a porous surface on top where it interfaces with cartilage, and a porous structure underneath with seemingly no transition from more dense subchondral bone to trabecular bone. (**F**) Bone volume/total volume (BV/TV) in the subchondral area (immediately underneath the cartilage) for aquatic mammals (blue) was significantly lower than in terrestrial mammals (green) (p<0.01).

The online version of this article includes the following figure supplement(s) for figure 4:

**Figure supplement 1.** Examples of the analysis of tissue from additional species.

**Figure supplement 2.** Lack of calcified cartilage in the sperm whale.

---

*1*, and additional images cartilage bone interface of the sperm whale provided in *Figure 4—figure supplement 2*). In fact, the cartilage-bone transition in aquatic mammals is a direct transition from hyaline cartilage to trabecular bone, without intermediate forms of both articular cartilage and subchondral bone, as is the case in terrestrial animals (*Figure 4D and E*). Bone volume over total volume (BV/TV) in the subchondral area (immediately underneath the cartilage-bone interface) averaged at 80.4 ± 16.8% for aquatic mammals, which was significantly lower than in terrestrial mammals (98.5 ± 0.8%, p<0.01, *Figure 4F*).

The trabecular bone below the subchondral plate under the central loading area of the humeri was denser, with thicker trabeculae in terrestrial mammals (34.5 ± 11.1% for terrestrial and 29.4 ± 10.3% for aquatic mammals) (*Figure 5A, B, C, D, E and F*). Trabecular thickness averaged at 211.7±141.1 μm in terrestrial mammals and at 120.0±27.9 μm in aquatic mammals. Trabecular BV/TV increased with size in terrestrial mammals ($R^2$=0.44, *Figure 5G*, green) and decreased with size in aquatic mammals ($R^2$=0.88, *Figure 5G*, blue). Thickness of the bone trabeculae (Tb.Th) increased with size in terrestrial mammals (*Figure 5H*, $R^2$=0.65, a=0.12), but was independent of size in aquatic mammals.

## Discussion

In the present study, it was investigated how osteochondral tissue has evolved differently in terrestrial and aquatic mammals to accommodate the different demands of life and locomotion on land or in water. Here, the characteristics of articular cartilage of aquatic mammals are described for the first time and they appeared to be widely different from all known terrestrial species in which this has been investigated. In contrast to the arcade-like (*Benninghoff, 1925*) collagen arrangement in

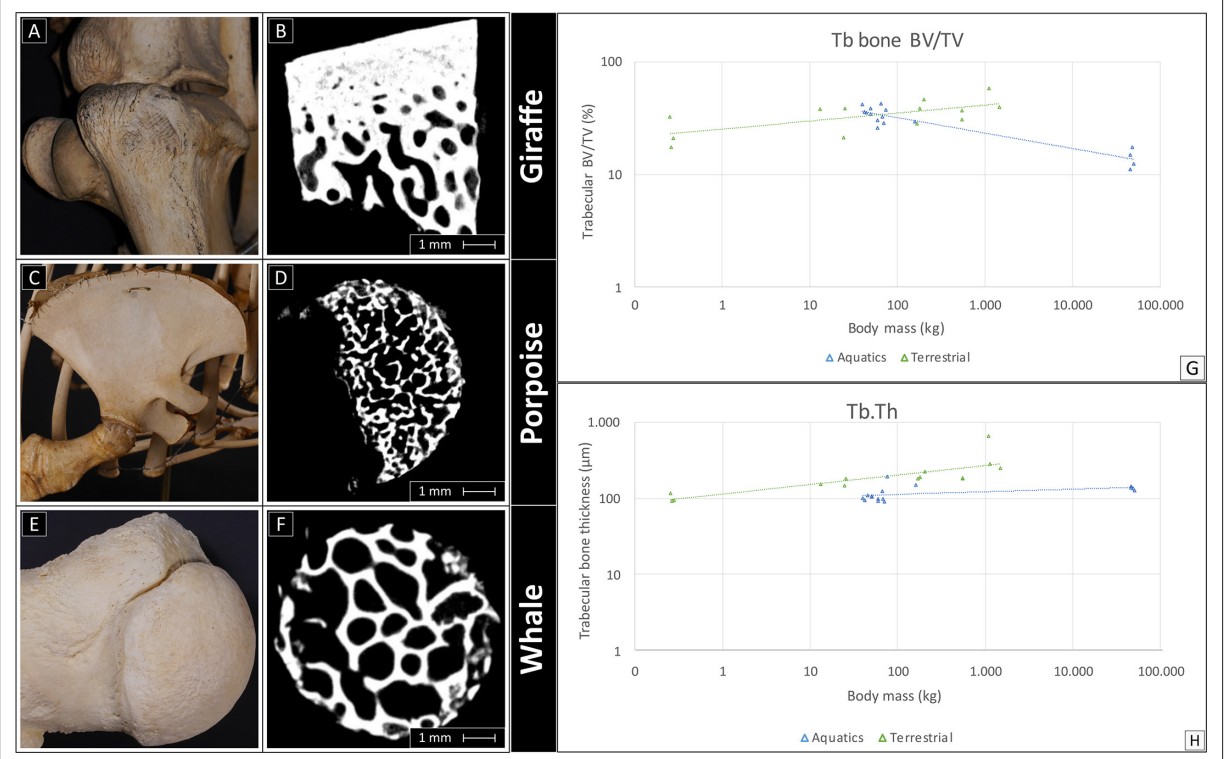

**Figure 5.** Comparison of the subchondral bone structures in the different species. Examples of macroscopic (left) and micro-CT scans (right) of a Rothschild's giraffe, porpoise and sperm whale. Left images show the macroscopic aspect of the humeral head of two aquatics and a terrestrial animal, on the right the correspondent micro-CT scan of the trabecular bone. (**A, B**) Terrestrial mammals featured a denser bone, while bone density was lower in harbor porpoises (**C, D**) and still lower in deep-diving animals (**E,F**). (**G**) Trabecular BV/TV increased with body mass in terrestrial mammals (green), and decreased with body mass in aquatic mammals (blue). (**H**) Trabecular bone thickness (Tb.Th) increased with size in terrestrial mammals (green), and was independent of size in aquatic mammals (blue).

the articular cartilage of terrestrial mammals, articular cartilage collagen fibers did not display this alignment in aquatic mammals. Moreover, the cartilage-bone interface of aquatic mammals entirely lacks the typical calcified layer present in all terrestrial species in which the osteochondral unit has been investigated. Further, it also lacks a dense subchondral plate, featuring an abrupt transition from hyaline cartilage to subchondral trabecular bone. Aquatic mammals lack the intermediate transitional forms of either cartilage (the calcified layer) or bone (the subchondral plate) that are typical of terrestrial species. Interestingly, for semi-aquatic species, such as the harbor seal (*Phoca vitulina*), there appears still to be some densification of the subchondral bone region, but this needs further investigation. Nevertheless, the trabeculae of the subchondral bone are thinner in aquatics than in terrestrials and they do not increase in size with increasing body mass, in contrast with what has been observed in terrestrial mammals.

Previously, it was shown that cartilage biochemistry is remarkably preserved among species, and that cartilage tissue displays microstructural features that allow functionality in mice as much as in elephants (*Malda et al., 2013*). One of the important features in the accommodation of different forces generated by body mass is the thickness of the tissue. Our group previously reported that articular cartilage thickness scaled with negative allometry in relationship to body mass (*a*=0.28), based on the analysis of nearly 60 terrestrial mammalian species (*Malda et al., 2013*). Interestingly, in the analysis presented here, it was observed that this correlation was similar in aquatic mammals, suggesting that this relationship might be dependent of factors other than loading, such as possibly metabolic limitations (*Holliday et al., 2010*). In hyaline cartilage, an avascular tissue in which nutrient supply is dependent on diffusion and fluid flow across a poroviscoelastic matrix (*Lu and Mow, 2008*), metabolite limitation is known to be related to tissue thickness (*Malda et al., 2013*; *White and Seymour, 2003*). In the sperm whale, an almost direct transition from hyaline cartilage to trabecular subchondral bone was observed, making diffusion from the rich vascular network of the trabecular

bone to the hyaline cartilage probably as easy as diffusion from the synovial fluid at the articular surface of that hyaline cartilage. The increased porosity, and thus the greater potential for transfer of solutes, may permit the great cartilage thickness seen in some aquatic mammals (over 7 mm in the largest whale [*Balaenoptera physalus*]). As an avascular tissue, cartilage is dependent on diffusion, so critical thickness will be larger if diffusion of nutrients can be provided more efficiently from two sides. The ratio of nutrient exchange is likely to be different in aquatic mammals, as at the few sites where the hyaline cartilage is in direct contact with the subchondral bone in terrestrial animals, solute exchange has been shown to be fivefold higher than through the calcified cartilage (*Arkill and Winlove, 2008*).

Simon et al. did not observe a consistent correlation between cartilage thickness and estimated compressive stress on the joint in a study conducted on five species of terrestrial mammals that varied widely in body mass (mouse, rat, dog, sheep, and cow; *Simon, 1970*). When studying the thicknesses of the different zones that make up articular cartilage, significant differences were observed between aquatic and terrestrial mammals in the deep and middle layers. However, this was not the case for the superficial layer. This supports the notion that the primary role of the superficial layer of cartilage is to protect the deeper layers from shear forces and to distribute impact forces and loads among the tissue (*Julkunen et al., 2007*; *Bevill et al., 2010*). Importantly, the load distribution of forces is essential in both aquatic and terrestrial species. Unlike shear forces, compressive loads will, however, be substantially lower in aquatic species, which would be a likely explanation for the relatively thinner deep layer.

A significant difference in the predominance of fiber orientation was found between aquatic and terrestrial mammals in all cartilage layers: collagen fiber orientation in terrestrial species followed the classic Benninghoff arcade model (*Benninghoff, 1925*). This is known to be conserved in all terrestrial species in which the configuration has been investigated (*Kääb et al., 1998*) and in which orientation is principally parallel to the surface in the superficial layer, and is changing in the middle zone as the fibers are bending towards an orientation perpendicular to the subchondral bone in the deep layer. The deep layer is known to increase in relative thickness with increasing body mass and therefore increasing compressive loading (*Felts and Spurrell, 1965*). Conversely, in aquatic mammals, orientation of the collagen fibers throughout the whole thickness of the cartilage was more random. In an aquatic mammal, the total biomechanical loading of the joints will be less because of the absence of gravity and in a relative sense, shear forces, which perhaps are less than in case of terrestrial locomotion, but are still generated by swimming (*Dejours, 1987*; *Williams and Worthy, 2009*; *Williams, 1999*; *Williams, 2001*), will become more important. This could potentially explain the presence of only a thin superficial layer. Moreover, it is likely that the heterogeneity in the alignment of the collagen fibers in aquatic mammals represents the strongly reduced need for a specifically organized orientation as seen in terrestrial mammals.

The different collagen fiber orientation observed in the aquatic mammals compared to terrestrial mammals is also reflected in the biomechanical behavior of cartilage. To analyze this effect, both peak and equilibrium modulus of cartilage were determined. These parameters are commonly used to describe the mechanical functionality of native and engineered cartilage because they provide a better understanding of how the constituents of cartilage contribute to its overall mechanical function (*Mow et al., 1984*; *Klika et al., 2016*). The mechanical responses observed in the aquatic mammals are indicative of an overall less stiff cartilage tissue, compared to that in terrestrial mammals. Furthermore, the absence of a defined collagen network structure in the aquatic mammals is likely to provide less resistance to osmotic swelling, as in terrestrial mammal's cartilage up to two thirds of the elastic modulus in compression arises from the electrostatic contributions and osmotic swelling that is restricted inside the organized collagen matrix (*Lai et al., 1991*; *Canal Guterl et al., 2010*).

The lack of a calcified cartilage layer and a compact subchondral plate, which are typical features of the osteochondral unit in terrestrial animals (*Sophia Fox et al., 2009*), was highly remarkable. The main function of the calcified layer in the osteochondral unit is to provide a gradual transition in stiffness from hyaline cartilage to subchondral bone and to serve as anchor point for the collagen fibers in the deep layer of the hyaline cartilage, that is to serve as a foundation for the Benninghoff arcades (*Sophia Fox et al., 2009*; *Benninghoff, 1925*; *Madry et al., 2010*). The subchondral plate has a role in distributing the compressive forces acting on the joint. As discussed earlier, in aquatic mammals compressive loading is hardly present and it may well be that firm anchoring of the collagen fibrils is

much less important for this reason. Detailed studies using high resolution electron microscopy might shed some light on this issue.

Bone density deeper under the cartilage tissue (trabecular BV/TV) was relatively higher in terrestrial mammals and increased with body mass, while the opposite was observed in aquatic mammals. For the terrestrial animals the relationship, this is in accordance with the earlier observation in these animals trabecular bone increases with increasing in size (*Barak et al., 2013*). The second observation, can be attributed to the lack of loading and the fact that in aquatic mammals living in deep waters, in which the absence of gravity obviates the need for such adaptation to increasing body mass, it has been shown that low bone density enables dynamic buoyancy control (*Gray et al., 2007*), explaining the opposite tendency in terrestrials. According to Dumont et al., bone microanatomy carries an ecological signal that holds information about habitat and locomotion patterns, which is particularly evident in tetrapods that have developed advanced secondary adaptations to life in water (*Dumont et al., 2013*). In the matter of buoyancy control of aquatic mammals, species recently adapted to aquatic life or living in shallow waters, such as sea cows (sirenians), have bones that are denser than in terrestrial forms to allow static control (*ballast*), as opposed to a lower bone density typically associated with the dynamic buoyancy control required by animals that exhibit deep diving behavior for longer periods (*Gray et al., 2007*; *Wall, 1983*).

Moreover, bone trabecular thickness behaved in a similar manner. This parameter also increased with size in terrestrial mammals, with a trend similar to previous reports in literature (*Mancini et al., 2019*; *Doube et al., 2011*), while in aquatic mammals trabecular thickness appeared to scale independently. This independence can possibly be explained by the lack of compressive forces necessary to elicit a proportionate response in the bone for trabecular thickness as suggested by Wolff's law (*Wolff, 1893*; *Frost, 2001*). In fact, Rolvien et al. investigated characteristics of the vertebral bodies of three toothed whale species sperm whale, orca (*Orcinus orca*) and harbor porpoise (*Phocoena phocoena*) and found that bone volume fraction (BV/TV) did not scale with body mass, but that trabeculae were thicker and fewer in number in larger whale species (*Rolvien et al., 2017*). Although research has shown the relationship between bone microanatomical organization, habitats and locomotion patterns in different taxa (*Dumont et al., 2013*), this was less clear when comparing different species of whales, particularly in view of the variability of for example diving-depth behavior of the different animals (*Rolvien et al., 2017*).

Interestingly, the histological sections of the aquatic mammals, displayed small islands of cartilaginous tissue embedded within the subchondral bone, and in the larger aquatic species, such as the sperm whale, large islands of cartilage tissue were found deep in the trabecular bone. These appeared as large lacunae upon first inspection with micro-CT, and an initial explanation was that these were a possible areas of osteocyte death due to barotraumas, which have been anecdotally described in deep-diving mammals (*Moore and Early, 2004*). However, upon histological evaluation, these islands appeared to be composed of healthy cartilage-like tissue naturally embedded in the bone, devoid of evident signs of bone remodeling, not suggesting any form of pathology. The areas may, therefore, represent incomplete osteochondral ossification, well-known in the horse and other species as osteochondrosis (*Thomas and Barnes, 2015*; *McCoy et al., 2013*) similarly to what was found and described by Rolvien et al. in the central areas of the vertebrae of sperm whales (*Rolvien et al., 2017*).

The number of individuals and species from which samples could be obtained was based on species availability and practical (such as the possibility of tissue harvest shortly after death from aquatic mammals) and does not compare to the numbers of earlier multispecies investigations in terrestrial animals (*Mancini et al., 2019*; *Malda et al., 2013*), which could be considered as a major limitation of this study. However, the included aquatic species covered a wide range of body masses (from 54 to 48,000 kg) and can, given the high consistency of the findings and the clear and unambiguous landslide differences with terrestrial species, be considered representative, especially in view of the very limited and subtle differences observed between a large number of different terrestrial species in the earlier studies (*Mancini et al., 2019*; *Malda et al., 2013*). Taken together, the findings of the current study add significantly to our knowledge of the basic aspects of the biology of the osteochondral unit and clearly reflect the crucial influence of loading. We know from evolution that life started to develop in the water and that later species came on land to develop further (*Colbert and Morales, 2001*). We also know that all mammals have initially evolved on land and that some of them returned to the water to become what we now call aquatic mammals (*Colbert and Morales, 2001*). Where we do not have

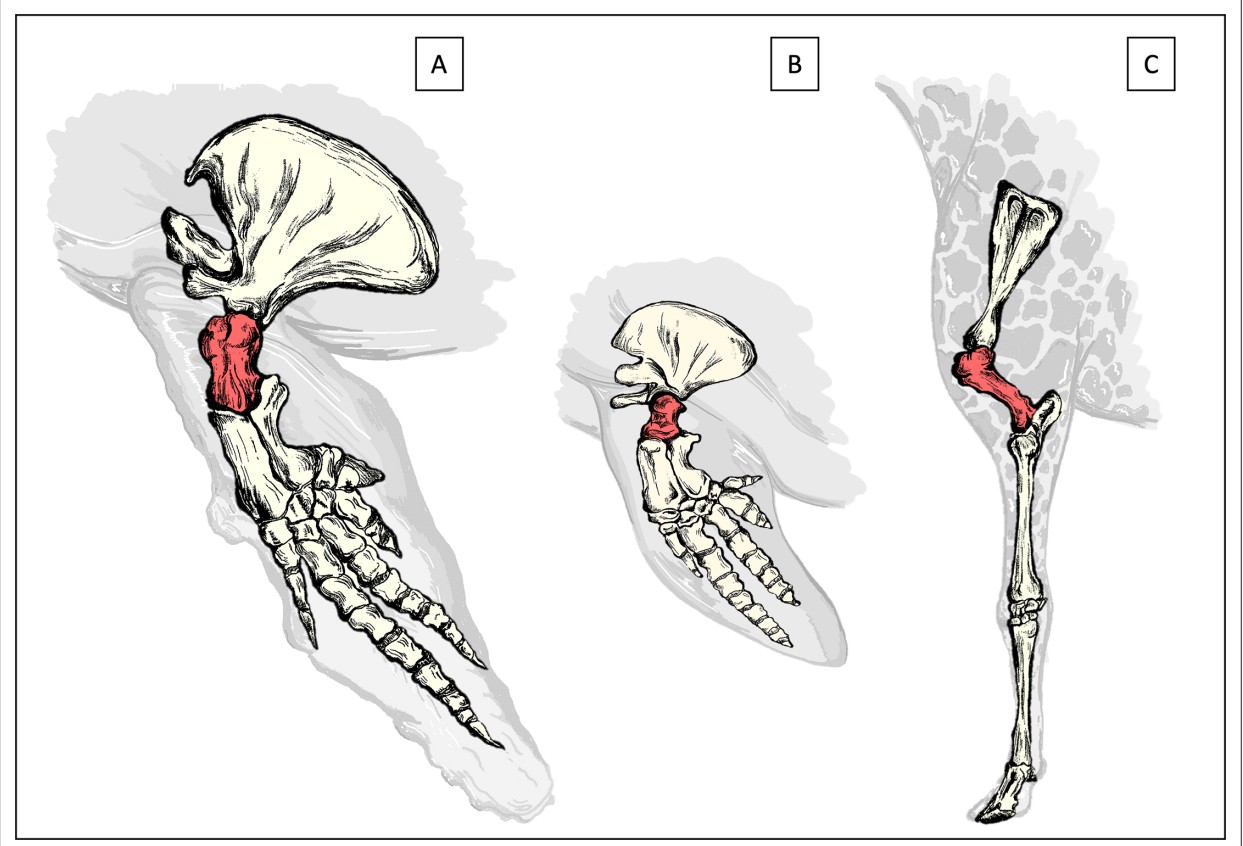

**Figure 6.** Schematic drawings of the front extremities. Drawings highlighting the location of the humerus (in red) of which the head (top) was sampled in this study for the sperm whale (**A**), harbor porpoise (**B**) and Rothschild's giraffe (**C**).

histological data from the joints of the land-borne species from which our current aquatic mammalian species have evolved, it is reasonable to assume that their osteochondral units were similar to the common format of all currently existing terrestrial mammalian species. The huge structural differences between the osteochondral units of the currently living aquatic and terrestrial mammals make hence clear that accommodating loading is the most important evolutionary pressure on the osteochondral unit of terrestrial mammals. Further, it highlights that this challenge is dealt with by the architecture and structural composition of the osteochondral unit and not by molecular composition or cellular diversity. The last two elements of the osteochondral unit are similar in aquatics and terrestrials. It is the specific tissue architecture and structural organization of especially the cartilage-bone interface that got lost in aquatic mammals, once the loading-based evolutionary pressure disappeared. This increased insight in the fundamental biology of the structural components of the osteochondral unit, obtained by observing Nature, is of great importance from a tissue repair perspective, as it is a natural guide for which considerations should be prioritized in the quest for recapitulation of the native template for regenerative purposes. In this sense, the study presents strong support for a recent call to rethink the current paradigms in articular cartilage regeneration (*Malda et al., 2019*), through focusing on solutions with a proper architecture that allow restoration of (biomechanical) function, rather than on mere cellular activity in terms of production of extracellular matrix components.

## Methods
### Collection of materials and tissues

To investigate the morphology of cartilage tissue and of the interface between cartilage and bone, histological analysis was performed on osteochondral samples taken from fresh humeral heads (*Figure 6*) of a variety of terrestrial and aquatic mammals (*Table 1*). Osteochondral tissue samples

**Table 1.** List of species included in the study.

In total tissue samples from 85 animals (34 for histological analysis, 29 for micro-CT and 20 for biomechanical analysis) were harvested for a total of 15 different species.

| | Species | Average body mass (kg) | Thickness | Histology (n) | Micro-CT (n) | Bio-mechanics (n) |
|---|---|---|---|---|---|---|
| 1 | Rat (*Rattus sp*) | 0.26 | 0.21 | 4 | 3 | - |
| 2 | European Badger (*Meles meles*) | 13.2 | 0.84 | 1 | 1 | 1 |
| 3 | Tufted deer (*Elaphodus cephalophus*) | 23 | 1.30 | 1 | - | - |
| 4 | Indian crested porcupine (*Hystrix indica*) | 24.5 | 0.5 | 1 | 1 | - |
| 5 | Cheetah (*Acinonyx jubatus*) | 25.5 | 0.48 | 1 | 1 | 2 |
| 6 | Harbor porpoise (*Phocoena phocoena*) | 54.5 | 1.11 | 9 | 9 | 9 |
| 7 | Harbor seal (*Phoca vitulina*) | 67.5 | 1.77 | 2 | 2 | 2 |
| 8 | Shetland pony (*Equus ferus caballus*) | 175 | 1.17 | 2 | 2 | - |
| 9 | Striped dolphin (*Stenella coeruleoalba*) | 180 | 1.37 | 1 | 1 | 1 |
| 10 | Onager (*Equus hemionus*) | 203 | 0.89 | 1 | 1 | - |
| 8b | Horse (*Equus ferus caballus*) | 550 | 1.38 | 2 | 2 | - |
| 11 | Rothschild's giraffe (*Giraffa camelopardalis*) | 1070 | 1.85 | 2 | 1 | 2 |
| 12 | White rhinoceros (*Ceratotherium simum*) | 1475 | 2.16 | 1 | 1 | - |
| 13 | Common minke whale (*Balaenoptera acutorostrata*) | 5100 | 5.05 | 2 | - | - |
| 14 | Sperm whale (*Physeter macrocephalus*) | 47.300 | 6.51 | 3 | 3 | 3 |
| 15 | Fin whale (*Balaenoptera physalus*) | 48.000 | 7.53 | 1 | 1 | 1 |
| | Total | | | 34 | 29 | 20 |

were harvested postmortem from the weight bearing central area of the humeral head of adult animals sent for necropsy to the division of Pathology, Faculty of Veterinary Medicine, Utrecht University, in the Netherlands. For aquatic large species, samples were harvested on site directly from animals that were found dead on the coasts of the Netherlands.

Animal species, age and body mass were recorded (or estimated in the case of whales based on species and body size), and macroscopic pictures of the joints were taken. Joints demonstrating

**Table 2.** Overview of cartilage samples on which biomechanical tests were conducted, with average diameter and thickness per species.

| Species | Number of animals | Number of samples | Cartilage diameter (mm) | Cartilage thickness (mm) |
|---|---|---|---|---|
| European Badger (*Meles meles*) | 1 | 1 | 6.4 | 1.7 |
| Cheetah (*Acinonyx jubatus*) | 1 | 2 | 6.5 | 2.1 |
| Rothschild's giraffe (*Giraffa camelopardalis*) | 2 | 3 | 6.9 | 1.7 |
| Harbor Porpoise (*Phocoena phocoena*) | 9 | 9 | 6.6 | 1.7 |
| Harbor Seal (*Phoca vitulina*) | 2 | 2 | 7.4 | 3.7 |
| Striped dolphin (*Stenella coeruleoalba*) | 1 | 3 | 7.6 | 1.9 |
| Sperm whale (*Physeter macrocephalus*) | 3 | 4 | 6.3 | 8.5 |
| Fin whale (*Balaenoptera physalus*) | 1 | 2 | 6.4 | 8.9 |

macroscopic or microscopic signs of cartilage degeneration were excluded; animals displaying signs of incomplete endochondral ossification were identified as skeletally immature and excluded as well.

In total 84 tissue samples (34 for histology and polarized light microscopy, 30 for micro-CT and 20 for biomechanical testing) were harvested from the humeral heads of mammals belonging to 15 different species, 9 terrestrial and 6 aquatic (*Table 2*); samples for histology were fixed in formalin 4%, while samples for micro-CT analysis were stored in 70% ethanol; samples for biomechanical testing were frozen immediately after harvest in Tissue-tek (Sakura Finitek, USA) until use.

## Histological preparation and analysis

Samples were fixed using 4% formalin, decalcified with Formical-2000 (EDTA/formic acid; Decal Chemical Corporation, Tallman, NY), dehydrated, cleared in xylene, embedded in paraffin and cut with a microtome to yield 5 μm sections. Sections were stained with fast green and Safranin-O for measurements of cartilage thickness (distance from the surface to the interface with the subchondral bone). Average thickness of cartilage tissue for each sample was determined by averaging 3 measurements per image taken from different locations of the section. Digital images were analyzed using ImageJ software (*Schindelin et al., 2012*).

## Polarized light microscopy

To evaluate the orientation of the cartilaginous collagen network, Polarized Light Microscopy (PLM) was chosen for its capacity to visualize the orientation of anisotropic materials (*Changoor et al., 2011*; *Yarker et al., 1983*; *Speer and Dahners, 1979*). Histological sections were stained with picrosirius red to stain the collagen fibers of the cartilage extracellular matrix (*Junqueira et al., 1979*). All microscopic images were acquired through an Olympus DP73 digital camera using Cell^F software (Matrix Optics, Malaysia). A combination of U-ANT and U-POT filters mounted on a light microscope (Olympus BX51, Olympus) was used for the PLM measurements. The two cross polarizers were used so that highly ordered collagen fibers that were perpendicular or tangential to the articular surface appeared bright or red, while fibers with other orientations (non-birefringent) appeared darkest. Collagen fiber orientation visualized via PLM was used for the classification of the superficial, middle and deep layer of cartilage (*Changoor et al., 2011*; *Rieppo et al., 2008*). The area with the minimum birefringence value was identified as the border between the superficial and the middle zones, whereas the deep zone was considered to begin when the orientation angle values reached a plateau (typically close to 90 degrees with respect to cartilage surface) (*Julkunen et al., 2007*; *Arokoski et al., 1996*). Micrographs acquired via PLM were converted to 8-bit images and the angle of orientation of the collagen structures was calculated within each of the three zones via automated image analysis and Fourier spectrum analysis, using the Directionality plug-in from the FIJI software (*Schindelin et al., 2012*). Data were plotted as histograms of frequency for each representative angle, and as radial plots (0° represent fibers parallel to the articulating surface) (*Figure 7*). For each of the histograms, the peaks centered at 0° and ±90° and were integrated to calculate their area using the OriginPro 8 software

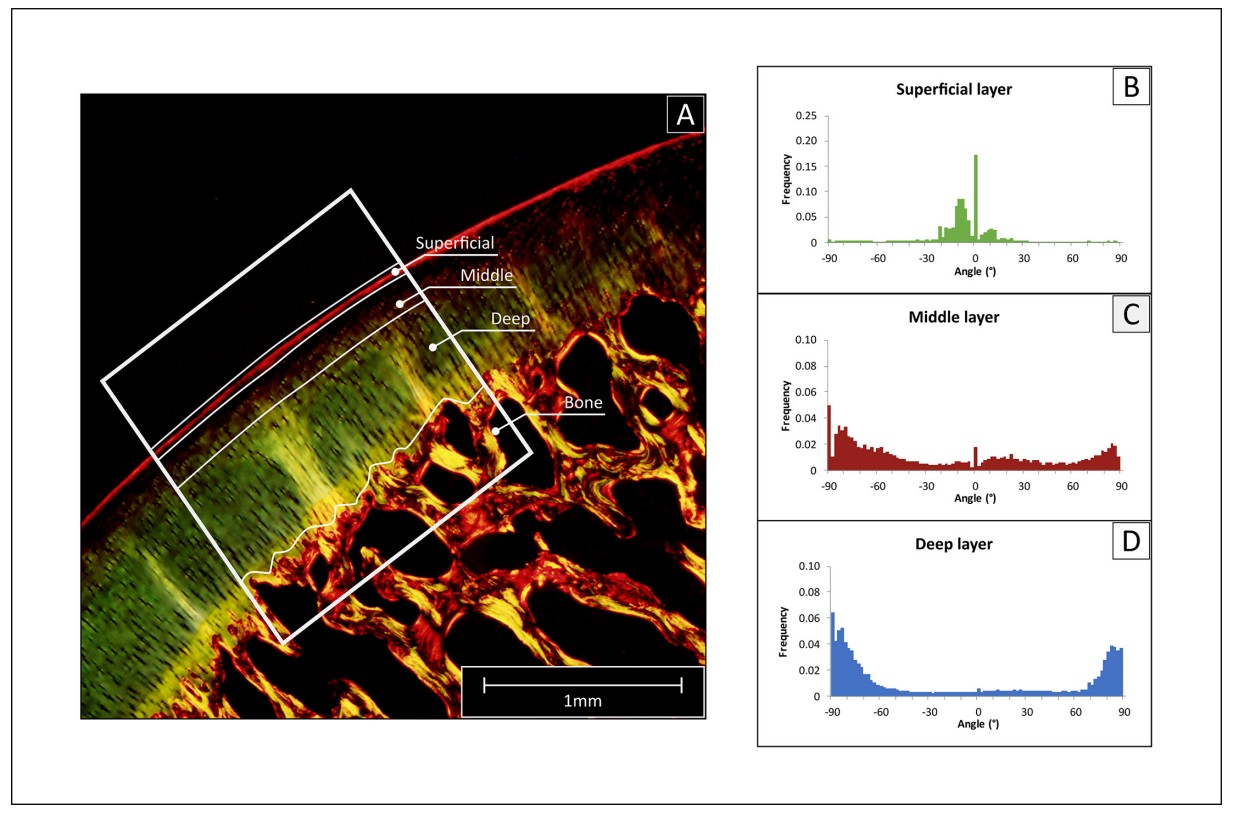

**Figure 7.** Polarized light microscopy (PLM) analysis of collagen fiber orientation. (**A**) The histological sections stained with picrosirius red were imaged, then a region of interest was selected (ROI, white frame), and the cartilage tissue was divided into deep, middle and superficial layer. (**B–D**) The selected region was analyzed with Image J to determine the frequency of fibers for angle increments of 2°, obtaining a detailed histogram for each region.

package (OriginLab, USA). Finally, cartilage relative layer thicknesses (%) was calculated based on the data obtained with PLM.

## Biomechanical testing and data analysis

The mechanical properties of the native cartilage of the osteochondral plugs of the different species were assessed by uniaxial unconfined compression using a universal testing machine (Zwick Z010, Germany) equipped with a 20 N load cell. Stress relaxation tests were performed by first applying a pre-load of 0.01 N to test samples and then strained to 15% at a rate of 10 μm/s, followed by a relaxation period of 900 s. Tests were conducted on cylindrical osteochondral plugs (*Table 2*). Prior to testing, samples were fixed to the bottom of a custom-made polycarbonate container using a cyanoacrylate-based adhesive and all tests were performed in PBS to approximate physiological conditions.

Peak and equilibrium stresses were calculated from each engineered stress-strain relaxation curve at peak or equilibrium, respectively. Here, stress is defined as the force divided by the specimen's unloaded cross-sectional area, and strain as the ratio between the original sample thickness and the displacement of compression platen. To quantify the relaxation response, a piecewise exponential function was fitted to the obtained stress-time curves, following the method by *Castilho et al., 2019*. Briefly, the fitted curves consist of three segments corresponding to an initial loading phase, and a fast and a slow relaxation phase. The formulae for the exponential curves for the fast and slow relaxation phases used are as follows,

$$\sigma(t) = \begin{cases} A_1 e^{-t/\tau_1} + B_1, & t < 100 \\ \quad . \\ A_2 e^{-t/\tau_2} + B_2, & t \geq 100 \end{cases} \qquad (1)$$

where the coefficients A and B are statistical parameters quantifying the shape of the relaxation curves and $\tau$ is a time constant that determines the rate of stress relaxation. The fitting procedure used minimizes the root mean square error between these curves and the measured stress, subject to the constraints that the fitted curves match the peak and equilibrium stress and that the two curves match at t=100 s.

## Micro-Computed Tomography

For the detailed analysis of the microstructural features of the subchondral and trabecular bone, Micro-Computed Tomography (micro-CT) was selected for the accurate measurement of micron-sized structures that constitute the bony tissue (*Fajardo and Müller, 2001*; *Holdsworth and Thornton, 2002*). Micro-CT images of the osteochondral cores were obtained with a micro-CT scanner (Quantum FX, Perkin Elmer, USA, voxel size = 20 µm³). The automatically reconstructed micro-CT images were subsequently converted to series of 2D TIFF images and were binarized using local thresholding (Bernsen technique). BoneJ software (*Doube et al., 2010*) was used to determine the trabecular thickness (Tb.Th) and the bone volume fraction (Tb BV/TV) of the bone immediately underneath the cartilage (subchondral bone; determined as previously described *Mancini et al., 2019*), and at the center of the bone of the osteochondral cores (trabecular bone).

## Statistical analysis

Comparisons between multiple groups were performed using a one-way ANOVA combined with post-hoc t-tests with Bonferroni correction. A one-way ANOVA with Tukey's post hoc test was used for comparison in biomechanical analysis. A regression analysis using a power curve fit was performed for correlations between body mass and the parameters of cartilage thickness, peak modulus, trabecular thickness, and trabecular BV/TV. When applied, normality and homogeneity were checked with Shapiro-Wilks and Levene's tests. The level of statistical significance was set at $p<0.05$. Statistical analysis was performed using GraphPad Prism 8 (GraphPad Software, Dotmatics, USA).

## Acknowledgements

The authors acknowledge the staff of the division of Pathology, from the Department of Biomolecular Health Sciences, of the faculty of Veterinary Medicine, in particular R Wagensveld-van den Dikkenberg, for their valuable help with the sample collection. The authors would also like to thank P Kamminga and the Naturalis Biodiversity Center in Leiden, the Netherlands, for access to their collection of mammalian skeletons. Post-mortem investigations of cetaceans in the Netherlands are commissioned by the Dutch Ministry of Nature, Agriculture and Food Quality (project grant numbers: 140000353; WOT04-009-045, HD3611/BO11018.02 065). The research leading to these results has received funding from the European Community's Seventh Framework Programme (FP7/2007– 2013) under grant agreement 309962 (HydroZONES), and the Dutch Arthritis Association (ReumaNL, LLP-12 and LLP-22).

## Additional information

### Funding

| Funder | Grant reference number | Author |
|---|---|---|
| European Commission | 3099622 (FP7) | Irina AD Mancini<br>P René van Weeren<br>Jos Malda |
| Dutch Arthritis Society | LLP12 | Riccardo Levato<br>Miguel Dias Castilho<br>P René van Weeren<br>Jos Malda |

| Funder | Grant reference number | Author |
| --- | --- | --- |
| Dutch Arthritis Society | LLP22 | Riccardo Levato<br>Miguel Dias Castilho<br>P René van Weeren<br>Jos Malda |
| Dutch Ministry of Nature, Agriculture and Food Quality | 140000353 | Lonneke L IJsseldijk |
| Dutch Ministry of Nature, Agriculture and Food Quality | WOT04-009-045 | Lonneke L IJsseldijk |
| Dutch Ministry of Nature, Agriculture and Food Quality | HD3611/BO11018.02 065 | Lonneke L IJsseldijk |
| European Community's Seventh Framework Programme | FP7/2007– 2013 (HydroZONES) | P René van Weeren |

The funders had no role in study design, data collection and interpretation, or the decision to submit the work for publication.

## Author contributions

Irina AD Mancini, Data curation, Formal analysis, Investigation, Visualization, Methodology, Writing – original draft, Writing – review and editing; Riccardo Levato, Data curation, Formal analysis, Supervision, Methodology, Writing – review and editing; Marlena M Ksiezarczyk, Jos Malda, Conceptualization, Resources, Data curation, Formal analysis, Supervision, Funding acquisition, Investigation, Visualization, Methodology, Writing – original draft, Project administration, Writing – review and editing; Miguel Dias Castilho, Data curation, Investigation, Methodology, Writing – review and editing; Michael Chen, Data curation, Formal analysis, Investigation, Methodology, Writing – review and editing; Mattie HP van Rijen, Data curation, Investigation, Visualization, Methodology, Writing – review and editing; Lonneke L IJsseldijk, Resources, Data curation, Formal analysis, Investigation, Methodology, Writing – review and editing; Marja Kik, Funding acquisition, Investigation, Methodology, Writing – review and editing; P René van Weeren, Conceptualization, Resources, Formal analysis, Supervision, Funding acquisition, Investigation, Methodology, Writing – original draft, Writing – review and editing

## Author ORCIDs

Lonneke L IJsseldijk ⓘ http://orcid.org/0000-0001-7288-9118
P René van Weeren ⓘ https://orcid.org/0000-0002-6654-1817
Jos Malda ⓘ http://orcid.org/0000-0002-9241-7676

## Decision letter and Author response

Decision letter https://doi.org/10.7554/eLife.80936.sa1
Author response https://doi.org/10.7554/eLife.80936.sa2

# Additional files

## Supplementary files

- MDAR checklist

## Data availability

Data generated or analysed during this study are included in the manuscript and supporting file; Source Data files have been provided for Figure 1.

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
