## [Editor Report]

It is important to determine microstructure-function relationships among different animal species, such as terrestrial and aquatic mammals, since it will help us understand articular biology and inform disease treatment. The authors compared the microstructure and biomechanical property of the osteochondral bone of the humeral head and found the cartilage of aquatic animals have a less stiff cartilage, a more random alignment of collagen fibers, and a lack of a calcified cartilage layer at the cartilage-bone interface. The specific composition of the osteochondral bone in aquatic mammals also reflects the changes in loading.

---

## [Decision Letter]

**Decision letter after peer review:**

Thank you for submitting your article "Nature's design of the osteochondral unit: microstructural differences between terrestrial and aquatic mammals" for consideration by *eLife*. Your article has been reviewed by 2 peer reviewers, and the evaluation has been overseen by a Reviewing Editor and Mone Zaidi as the Senior Editor. The following individual involved in the review of your submission has agreed to reveal their identity: Magali Cucchiarini (Reviewer #2).

Essential revisions:

1) An additional figure plate showing a selection of further species, especially the high extremes would further increase the value of the manuscript.

2) Making schematic drawings that indicate the location of the humerus joint for the three most presented species will help readers understand the anatomical and biomechanical context.

3) In the legend of Figure 4, the authors stated that terrestrial mammals display a tide mark, implying that aquatic animals do not have a tide mark. However, the data presented in Figure 4A showed that aquatic animals seem to have a tide mark. The authors need to explain this discrepancy.

*Reviewer #1 (Recommendations for the authors):*

Within this comparative study of terrestrial and aquatic mammals using morphological and biomechanical methods, the authors could point out differences that most probably relate to their movement under buoyancy versus full gravity. Most importantly aquatic animals have a less stiff cartilage (at equilibrium), a more random alignment of collagen, and a lack of calcified cartilage leading to a continuous transition between cartilage and bone. Uniaxial compression tests, polarization microscopy, histochemistry, and µCT provided the necessary information about stress relaxation, fiber orientation, and cartilage versus bone localization. Despite the limited number of species presented in more detail, the results were sound and conclusive and elucidated new aspects of cartilage extension into the subchondral space which is interesting for developmental biology as well as tissue regeneration.

Manuscript in general

The manuscript is well structured and written and data are presented thoroughly. The methods used are appropriate to answer the research questions and histological, polarization microscopic, and µCD images are of high quality.

Despite the high number of species included in the study, only three species were presented in more detail. An additional figure plate showing a selection of further species, especially the high extremes (in size, etc.) would further increase the value of the manuscript.

In order to demonstrate the different anatomical situations of the three most presented species (sperm whale/porpoise and giraffe) authors could add schematic drawings indicating the location of the humerus joint for easier understanding of the anatomical and biomechanical context.

Introduction:

Introduction 3rd paragraph: "The function of synovial joints in particular, is to minimize frictions between those bony components by providing lubrication…" is not perfectly accurate since the function of synovial joints is to articulate. The minimization of the friction is given by its anatomy and in particular the cartilage.

Results:

Use consisting nomenclature in e.g. Figure 2 A: Whale B Porpoise – porpoise is also a whale, so rather use sperm whale also in the image.

Figure 1 A, B: the informative value of the figure could further be increased by declaring which data point belongs to which species e.g. by replacing triangles with numbers and adding these numbers to Table 1 also.

Figure 1C: a box blot might increase the information of the displayed graph by not only displaying the upper limit but the range of the relative thickness of the three cartilage zones.

Important point: Polarization microscopy of picro Sirius red-stained sections not only provides information about the fiber orientation but also on fiber thickness: small fibers are green-yellow; large fibers are orange-red. In Figure 2. A-C the aquatic animals show green polarization in the deep zone and red polarization in the main part while in the giraffe the main lower part appeared yellow. Is this conformable with other aquatic and terrestrial species?

Important point: Furthermore, in Figure 2A the main part of the fibers appears oriented parallel to the joint surface and not without any predominant orientation as described in the text. Please comment on that or change the text accordingly.

Figure 2 D, E Are the orientation and average peak areas for the three species in Figure A-C or for all species included in this study?

Figure 2 D: Please explain the x-axis in the polar graphical representation.

Figure 2 E: Box plot might again be more informative.

Figure 3A: Is it possible to show also the curves from the other species that were measured?

3E, F: please declare which data point belongs to which species.

Important point: Figure 4 Figure legend 4C says that terrestrial mammals display a tide mark. Does this formulation seem to implicate that aquatic animals do not have a tide mark? However, in the insert of Figure 4A, a line is visible which also looks like a tide mark. Can you please comment on that?

Figure 4D: The legend says that the figure shows on top a dense surface area that interfaces with the (calcified) cartilage (not shown in the scan). How is this meant? Was the calcified cartilage removed?

Discussion:

"articular cartilage collagen fibers did not display a preferential alignment in aquatic mammals." Figure 2A appears as if the main part (middle layer) was aligned horizontally (parallel to the surface).

"…This supports the notion that the primary role of the superficial layer of cartilage is to distribute impact forces and loads among the tissue" In addition, the superficial zone (thickness and structure) is a consequence of the shear force which influences the tissue mainly to a restricted depth. This might also be a predominant reason for the low thickness of the superficial zone.

"A significant difference in the predominance of fiber orientation was found between aquatic and terrestrial mammals in all cartilage layers" – except in the superficial zone?

"In the matter of buoyancy control of aquatic mammals, species recently adapted to aquatic life or living in shallow waters …" maybe add an example of such a species.

Important point: For the mechanical test osteochondral cylinders were used. Could the authors please explain in more detail about the standardization and the proportion of bone and cartilage (e.g. total thickness of the plugs always the same)? In case the bone was of different thicknesses, how do the authors think that it could influence the values of the measurements and therefore the interpretation of the data?

The large islands of cartilage tissue within the trabecular bone are very interesting. Do the authors have any explanation for that? Could it be residuals of endochondral ossification or be related to an increased cross-talk between cartilage and bone due to a more continuous transition and less complete separation of bone and cartilage?

Materials and methods:

Were micro CT samples subsequently used for histology?

For completeness please mention the program that was used for statistics.

---

## [Author Response]

Essential revisions:1) An additional figure plate showing a selection of further species, especially the high extremes would further increase the value of the manuscript.

This has now been included as Figure 4—figure supplement 1.

2) Making schematic drawings that indicate the location of the humerus joint for the three most presented species will help readers understand the anatomical and biomechanical context.

This has now been included as Figure 6.

3) In the legend of Figure 4, the authors stated that terrestrial mammals display a tide mark, implying that aquatic animals do not have a tide mark. However, the data presented in Figure 4A showed that aquatic animals seem to have a tide mark. The authors need to explain this discrepancy.

This discrepancy is now further discussed below in the response to the issue raised by Reviewer 1 and an additional figure is added that includes higher magnification of the sections of the tissue obtained from three different sperm whales (Figure 4—figure supplement 2).

Reviewer #1 (Recommendations for the authors):Within this comparative study of terrestrial and aquatic mammals using morphological and biomechanical methods, the authors could point out differences that most probably relate to their movement under buoyancy versus full gravity. Most importantly aquatic animals have a less stiff cartilage (at equilibrium), a more random alignment of collagen, and a lack of calcified cartilage leading to a continuous transition between cartilage and bone. Uniaxial compression tests, polarization microscopy, histochemistry, and µCT provided the necessary information about stress relaxation, fiber orientation, and cartilage versus bone localization. Despite the limited number of species presented in more detail, the results were sound and conclusive and elucidated new aspects of cartilage extension into the subchondral space which is interesting for developmental biology as well as tissue regeneration.Manuscript in generalThe manuscript is well structured and written and data are presented thoroughly. The methods used are appropriate to answer the research questions and histological, polarization microscopic, and µCD images are of high quality.Despite the high number of species included in the study, only three species were presented in more detail. An additional figure plate showing a selection of further species, especially the high extremes (in size, etc.) would further increase the value of the manuscript.

We thank the reviewer for this comment. Following this suggestion, we have now included a new supplementary figure (Figure 4—figure supplement 1), in which we included additional µCT’s and histological stainings (safranin-O/fast green) of additional species among the ones screened in this study. We now extended the selection to include data from rat, harbor seal and white rhinoceros tissues and included a reference to the figure in the manuscript text.

Further to the Discussion section the following was added:

“Interestingly, for semi-aquatic species, such as the harbor seal (Phoca vitulina), there appears still to be some densification of the subchondral bone region, but this needs further investigation.” (page 7)

In order to demonstrate the different anatomical situations of the three most presented species (sperm whale/porpoise and giraffe) authors could add schematic drawings indicating the location of the humerus joint for easier understanding of the anatomical and biomechanical context.

We thank the reviewer for pointing this out. We have included this figure in the revised manuscript (Figure 6).

Introduction:Introduction 3rd paragraph: "The function of synovial joints in particular, is to minimize frictions between those bony components by providing lubrication…" is not perfectly accurate since the function of synovial joints is to articulate. The minimization of the friction is given by its anatomy and in particular the cartilage.

Thank you for this comment. We have now adjusted the manuscript text accordingly:

"The function of synovial joints in particular, is to ensure the correct articulation of adjacent bones. In addition, it minimizes friction between the bony components by providing a smooth articulating surface and lubrication…" (page 3)

Results:Use consisting nomenclature in e.g. Figure 2 A: Whale B Porpoise – porpoise is also a whale, so rather use sperm whale also in the image.

We thank the reviewer for highlighting this. We have indeed simplified the labels in the figures, to minimize their visual impact on the figure. However, to clarify the point raised by the reviewer, and to ensure consistency and correctness of the manuscript, we have now systematically specified this in the all applicable figure legends.

Figure 1 A, B: the informative value of the figure could further be increased by declaring which data point belongs to which species e.g. by replacing triangles with numbers and adding these numbers to Table 1 also.

We agree with the reviewer that it would be informative if the specific thicknesses of the cartilage tissue in the individual species could be retrieved from the manuscript. We have now provided the average values for the different terrestrial and aquatic species in Figure 1B (which does slightly impact on the R2 and a values; this has also been adjusted in the revised manuscript text) To ensure overall clarity we have extended Table 1 with the summary data of the average total cartilage thickness per species.

Figure 1C: a box blot might increase the information of the displayed graph by not only displaying the upper limit but the range of the relative thickness of the three cartilage zones.

We do thank the reviewer for this suggestion and have changed Figure 1C accordingly.

Important point: Polarization microscopy of picro Sirius red-stained sections not only provides information about the fiber orientation but also on fiber thickness: small fibers are green-yellow; large fibers are orange-red. In Figure 2. A-C the aquatic animals show green polarization in the deep zone and red polarization in the main part while in the giraffe the main lower part appeared yellow. Is this conformable with other aquatic and terrestrial species?

The reviewer brings up an important point. It is known that in terrestrial mammals the collagen fiber thickness varies and increases with dept (see for example, Changoor *et al.* OAC 19:1458, 2011). However, visualization of picro Sirius red-stained samples cannot be directly related to the fiber thickness alone as the packing of the collagen molecules is an additional determining factor (Dayan *et al.* Histochem 93:27, 1989), which is of particular relevance in the densely packed cartilage tissue. Therefore, we have not commented on this in the revised manuscript.

Important point: Furthermore, in Figure 2A the main part of the fibers appears oriented parallel to the joint surface and not without any predominant orientation as described in the text. Please comment on that or change the text accordingly.

The reviewer raises an important point indeed. The statement in the manuscript text was quite strong. There is a clear distinction between the fiber orientation in the aquatic and terrestrial samples, however minor alignment can indeed be observed. We have adjusted it now to better reflect our observations. The text now reads:

“…whereas for the aquatic mammals this arcade orientation of fibers could not be distinguished (Figure 2D-E). However, some alignment in the superficial layer was also observed (Figure 2B), albeit not to the same extend as in the samples of terrestrial mammals” (page 6)

Figure 2 D, E Are the orientation and average peak areas for the three species in Figure A-C or for all species included in this study?

Indeed, the provided orientation and average peak areas reported are based on all samples (species) included in the study.

Figure 2 D: Please explain the x-axis in the polar graphical representation.

The information in the radial in the graph is the normalized frequency (how many features the software recognizes) per angle (in degrees), see also Figure 6 in the manuscript for further explanation.

Figure 2 E: Box plot might again be more informative.

We do thank the reviewer for this suggestion and have changed Figure 2E accordingly.

Figure 3A: Is it possible to show also the curves from the other species that were measured?3E, F: please declare which data point belongs to which species.

Inclusion of the additional curves did not improve the legibility of the graph and as the figure was only intended as an example we have not further altered the original figure.

Important point: Figure 4 Figure legend 4C says that terrestrial mammals display a tide mark. Does this formulation seem to implicate that aquatic animals do not have a tide mark? However, in the insert of Figure 4A, a line is visible which also looks like a tide mark. Can you please comment on that?

We thank the reviewer for pointing this out. The observed line, or dark border, on the image in Figure 4A is actually not the tidemark, but an artifact (wrinkle) in the slide as the large cartilage-bone samples are challenging to process due to the swelling behavior of the cartilage matrix. To support our claim and clarify this, we have included high magnification images of the bone cartilage interface of all three sperm whale samples included in this study as an additional supplementary figure (Figure 4—figure supplement 2).

Figure 4D: The legend says that the figure shows on top a dense surface area that interfaces with the (calcified) cartilage (not shown in the scan). How is this meant? Was the calcified cartilage removed?

We apologize for this unclarity. We meant that the cartilage layer is not visualized in the scan (though it was still present during the analysis). We have removed “(calcified)” from the legend text to make this more clear.

Discussion:"articular cartilage collagen fibers did not display a preferential alignment in aquatic mammals." Figure 2A appears as if the main part (middle layer) was aligned horizontally (parallel to the surface).

We agree with the reviewer that the statement is confusing and have adjusted it in the revised manuscript accordingly. It now reads:

“In contrast to the arcade-like (Benninghoff^23^) collagen arrangement in the articular cartilage of terrestrial mammals, articular cartilage collagen fibers did not display this alignment in aquatic mammals..”(page 7)

"…This supports the notion that the primary role of the superficial layer of cartilage is to distribute impact forces and loads among the tissue" In addition, the superficial zone (thickness and structure) is a consequence of the shear force which influences the tissue mainly to a restricted depth. This might also be a predominant reason for the low thickness of the superficial zone.

We thank the reviewer for bringing that up and we have included this in the discussion.

“This supports the notion that the primary role of the superficial layer of cartilage is to protect the deeper layers from shear forces and to distribute impact forces and loads among the tissue^29,30^. Importantly, the load distribution of forces is essential in both aquatic and terrestrial species. Unlike shear forces, compressive loads will, however, be substantially lower in aquatic species, which would be a likely explanation for the relatively thinner deep layer.” (page 9)

"A significant difference in the predominance of fiber orientation was found between aquatic and terrestrial mammals in all cartilage layers" – except in the superficial zone?

A significant different in predominance of fiber orientation was found for all layers, including the superficial zone as is underscored in the lower intensity in the Figures 2A and 2B, as well as the lower frequency reported in Figure 2D for the aquatic samples. However, a thin superficial layer was present. We have now changed the discussion to better reflect this.

“Conversely, in aquatic mammals, orientation of the collagen fibers throughout the whole thickness of the cartilage was more random. In an aquatic mammal, the total biomechanical loading of the joints will be less because of the absence of gravity and in a relative sense, shear forces, which perhaps are less than in case of terrestrial locomotion, but are still generated by swimming^6,32-34^, will become more important. This could potentially explain the presence of only a thin superficial layer.”

"In the matter of buoyancy control of aquatic mammals, species recently adapted to aquatic life or living in shallow waters …" maybe add an example of such a species.

We thank the reviewer for the suggestion and have now included an example of such an aquatic mammal that lives in shallow water (sea cow).

Important point: For the mechanical test osteochondral cylinders were used. Could the authors please explain in more detail about the standardization and the proportion of bone and cartilage (e.g. total thickness of the plugs always the same)? In case the bone was of different thicknesses, how do the authors think that it could influence the values of the measurements and therefore the interpretation of the data?

The overall stiffness of the bone is about three orders of magnitude higher than that of the cartilage. In view of this, we have assumed minimal effects of the bone component on the outcome of the testing.

The large islands of cartilage tissue within the trabecular bone are very interesting. Do the authors have any explanation for that? Could it be residuals of endochondral ossification or be related to an increased cross-talk between cartilage and bone due to a more continuous transition and less complete separation of bone and cartilage?

We do thank the reviewer for raising this point. Indeed, this could be remnants of the process of osteochondral ossification. One could actually postulate that due to the limited direct loading the developmental process does not need or is not induced to proceed further and hence does not result in a dense subchondral bone plate. In our clinical experience, we have seen these remnants in (young) horses as well. In those animals these islands may end up as necrotic cartilage (as the original vasculature of the growth cartilage disappears) that in the end may give rise to the formation of fissures and joint mice (loose fragments; osteochondrosis). However, for the aquatics, one can presume that, if the cartilage islands are not too large and are located in quite open and hence very richly vascularized trabecular bone *and are not loaded*, they might survive through diffusion of tissue fluid. However, this would need further investigation.

Materials and methods:Were micro CT samples subsequently used for histology?

No, separate samples were obtained for micro CT and for histology.

For completeness please mention the program that was used for statistics.

We apologize for omitting to include this and have now added this to the revised manuscript:

“Statistical analysis was performed using GraphPad Prism 8 (GraphPad Software, Dotmatics, USA).” (page 17)